# Algae: Critical Sources of Very Long-Chain Polyunsaturated Fatty Acids

**DOI:** 10.3390/biom9110708

**Published:** 2019-11-06

**Authors:** John L. Harwood

**Affiliations:** School of Biosciences, Cardiff University, Cardiff CF10 3AX, UK; Harwood@Cardiff.ac.uk

**Keywords:** essential fatty acids, human requirement, eicosapentaenoic acid (EPA), docosahexaenoic acid (DHA), fish lipids, algal lipid synthesis

## Abstract

Polyunsaturated fatty acids (PUFAs), which are divided into n-3 and n-6 classes, are essential for good health in humans and many animals. They are metabolised to lipid mediators, such as eicosanoids, resolvins and protectins. Increasing interest has been paid to the 20 or 22 carbon very long chain PUFAs, since these compounds can be used to form lipid mediators and, thus, avoid inefficient formation of dietary plant PUFAs. The ultimate sources of very long chain PUFAs are algae, which are consumed by fish and then by humans. In this review, I describe the biosynthesis of very long chain PUFAs by algae and how this synthesis can be manipulated for commercial purposes. Ultimately, the production of algal oils is critical for ecosystems worldwide, as well as for human dietary lipids.

## 1. Introduction

Polyunsaturated fatty acids (PUFAs) are essential components in the diet of humans [1,2] and many other animals. There are two classes, which belong to the n-3 and the n-6 families. The basic precursors of these two families are alpha-linolenic and linoleic acids, respectively, and their dietary essentiality was first recognised 90 years ago [3,4]. However, because the PUFAs give rise to lipid signalling molecules or mediators, usually formed of 20 or 22 carbons, but are poorly converted to such precursor PUFAs, there is considerable interest in dietary 20 or 22C PUFA (very long chain PUFAs: VLCPUFA). Indeed, under certain conditions, such fatty acids may be ‘conditionally essential’ [5]. Thus, the dietary content of fatty acids, such as arachidonic (ARA), eicosapentaenoic (EPA) and docosahexaenoic (DHA), acids is important.

Dietary PUFA can be metabolised in animals to yield 20C or 22C products by a series of desaturation and elongation reactions (Figure 1). For DHA in mammals the synthesis from EPA involves the ‘Sprecher pathway’ [1,6], with a 24C intermediate and chain shortening by beta-oxidation. However, this biosynthetic route has recently been re-examined [7]. Once 20 or 22C PUFA have been produced they can be subject to oxidation by cyclooxygenase, lipoxygenase or epoxidase enzymes (Figure 2). This will give rise to a host of signalling molecules, the balance of which will depend on the substrates available, as well as the activity of the oxidases themselves. As a generalisation, n-6 PUFA give rise to inflammatory mediators, while the n-3 PUFA form neutral or anti-inflammatory signalling molecules [1]. Thus, the balance of (dietary) n-3 versus n-6 PUFA is important in determining overall physiological effect and it is thought that many modern ‘Western diets’ contain an excess of n-6 PUFA constituents. This can result in a chronic inflammatory situation which is of relevance to important diseases, such as arthritis, dementia or cardiovascular disease [8,9,10]. Western diets may contain a ratio of 15–20/1 (n-6/n-3 PUFA), but a ratio of about 4:1 has been recommended [11].

Apart from their importance in giving rise to non-inflammatory eicosanoids, n-3 PUFA are also important in allowing the biosynthesis of some newly discovered lipid mediators [2,12,13,14,15]. Amongst these, resolvins, maresins and protectins are particularly important [1]. Resolvins act to resolve an acute inflammatory episode. Those derived from EPA are of the E series, while those from DHA are in the D series [1,16,17,18]. A single oxygenation found in human macrophages or platelets can give rise to a mediator termed maresin (macrophage mediator in resolving inflammation) [13]. Alternatively, protectins can be produced, of which the first to be discovered was neuroprotectin in brain tissue [19]. The production of resolvins, maresins and protectins from DHA finally gives a reason for the high amounts of DHA in neural tissue (see [11]).

Because of the inefficient production of EPA and DHA from alpha-linolenic acid (ALA) [20,21,22], the often overwhelming amount of linoleic acid (LA) in diets, and the increasing recognition of the importance of adequate n-3 PUFA for good health [11,20], the intake of dietary EPA and DHA has assumed considerable interest in recent years. Although fish are the main source of such acids for humans, EPA and DHA are produced de novo, mainly in algae. Thus, in recent years, there has been an accelerating interest in the biosynthesis of EPA and DHA in algae [23,24], and also in the use of genes from such algae to modify crop plants to produce ‘fish oils’ [25,26,27,28,29,30]. Moreover, many fish do not contain (or produce) significant amounts of EPA and DHA, and human dietary recommendations specify ‘oily’ fish as being necessary. When fish, such as salmon, are farmed, then fish meal or fish oil are needed to supplement the feed. Otherwise, the levels of EPA and DHA in the fish will be lowered dramatically [31]. Clearly, this is an unsustainable situation and emphasises the need to find alternative sources of VLCPUFA for fish farming—such as algae-derived additives.

As pointed out earlier, the 18C PUFAs, LA and ALA, are the basic essential fatty acids [3,4,5]. Although such acids can usually supply the basic human needs of VLCPUFA [5,31], there is a lot of evidence that dietary EPA and DHA have beneficial effects for good health. These benefits include reduced risk of widespread and important diseases, such as arthritis [32,33,34] and cardiovascular disease [35,36,37,38], and improved brain function. For the latter, three major epidemiological surveys have shown that significant amounts of n-3 PUFA in the diet can reduce the subsequent incidence of dementia [39,40,41]. Moreover, general brain function needs n-3 VLCPUFA [42] and supplementation can improve cognition in some patients [43,44]. In addition, a recent meta-analysis of various studies into nutrient supplements found that the strongest evidence for beneficial effects was for PUFAs (particularly EPA) as an adjunctive therapy for depression [45].

## 2. Fish as a Source of Very Long Chain Polyunsaturated Fatty Acids

While fish are a convenient (and usual) source of important lipids, such as the VLCPUFAs, many commercial fish (e.g., cod or tilapia) only contain small amounts of oil in their fillets. To satisfy human requirements, oily fish are needed. The fatty acid compositions of oils from important fish sources are shown in Table 1. It can be seen that n-3 VLCPUFA contents are in the range 14–31% of total fatty acids, with a wide-ranging EPA:DHA ratio. Because it seems clear that humans can interconvert these two fatty acids [7], it may not matter whether EPA or DHA is the dominant fish oil VLCPUFA. Nevertheless, because of the background of research and the known high concentrations of DHA in human (especially neurological) tissue, most attention has been paid to ensuring that DHA intakes are maintained. Although fish oils remain the most convenient source of n-3 VLCPUFA, there is increasing concern about the ability of this source to satisfy growing demand by humans, let alone their use in aquaculture. Indeed, most of the current supply of fish oils is used in aquaculture [46,47]. Moreover, there may be significant amounts of undesirable compounds (e.g., dioxins, mercury) in fish oils. Such hazardous contaminants can be removed [46], but this adds considerably to the manufacturing costs. These considerations emphasise the desirability of algae as a source of VLCPUFA.

While much of the recent research on the benefits of dietary PUFAs has concentrated on humans, such fatty acids are important for other animals. Gladyshev and Sushchik have reviewed ecosystems, with regard to VLCPUFA [48]. There are many examples where various animals (e.g., zooplankton, *Daphnia*) benefit from an adequate supply of appropriate PUFA from algae (e.g., [49,50,51,52]). Moreover, fish themselves need an appropriate intake of PUFA in their diet [47,53]. Thus, there are more and more examples where algae (or new genetically-manipulated agricultural crops) are needed to reduce the current and future projected shortfalls in VLCPUFA. The anticipated shortage of VLCPUFA will only be made worse by climate change [54].

Given the increasing demand for VLCPUFA, as well as the problems with maintaining fish supplies, it is clear that this is a serious issue, that needs to be addressed urgently. Some calculations about the amounts of EPA and DHA which will be needed in the future [48] and possible shortfalls due to climate change [54] have been made.

## 3. Algal Production of Polyunsaturated Fatty Acids

The major lipid classes in algae are membrane components (betaine ether lipids, glycosylglycerides, phosphoglycerides) and storage lipids (mainly triacylglycerols) [23,24]. Numerous other compounds are found in small amounts [23]. Depending on the alga, the proportions of the polar membrane lipids may vary considerably [24] but they all tend to have a high concentration of PUFA. Evaluation of the overall fatty acid composition of algae, in addition to their location in different acyl lipids, have been well summarised [23,24,55,56]. Moreover, some algae have been successfully used for ‘single cell oils’ [57].

Recently, our knowledge of algal fatty acid composition has been enhanced by the establishment of the SAG culture collection (Culture Collection at the University of Gottingen, Germany). The fatty acid composition of individual algae and their lines (as grown to stationary phase) have been surveyed [58]. Some examples, which emphasise not only the wide fatty acid composition of different algae, but also their variability even within the same class, are shown in Table 2. It should be noted that growth conditions, seasonal variations and developmental stages will all impact the overall fatty acid composition of algae, in addition to the patterns for individual lipid classes [23,55,56,59].

As discussed in previous reviews [24,25,55,56], certain classes of algae contain notable amounts of VLCPUFA. For example, brown algae are often enriched in arachidonate (ARA) [55], while red algae can have significant EPA. Sometimes, the EPA occurs with ARA (as in *C. crispus*) [55,60] or without (as in *Palmaria palmata*, where EPA can represent half of the total fatty acids) [60,61].

De novo production of fatty acids in eukaryotic algae begins with the activity of acetyl-CoA carboxylase and fatty acid synthase within the chloroplasts. Information about individual enzymes and their regulation has been based on higher plant systems [62], which was then supplemented by work with the model green alga *Chlamydomonas reinhardtii* [63,64]. Since then, further detailed examination of several algae, including the eustigmatophyte *Nannochloropsis* spp., the diatom *Phaeodactylum tricornutum* and the red alga, *Cyanidioschyzon merolae* [65,66,67,68,69,70,71], suggested that fatty acid synthesis is similar to that in *C. reinhardtii* and, hence, in higher plants [62,72,73,74]. Details of the enzymes involved in de novo fatty acid synthesis are provided in a recent review [24].

As in higher plants, algal fatty acid synthesis in chloroplasts results in 16 and 18-carbon products. Whole further desaturations can take place in plastids. Production of VLCPUFA usually requires a mixture of elongation and desaturation on the endoplasmic reticulum (ER). Thus, fatty acids have to move from the chloroplast through the cytosol to the ER. This process has been compared between algae and higher plants [72,75].

Production of VLCPUFA has been examined in detail in *Phaeodactylum tricornutum* [71], where high amounts of EPA are made (Table 2), as well as in several other species (see [24]). The conventional delta6-pathway begins with linoleic (LA) or alpha-linolenic (ALA) acids. This involves delta6-desaturase, delta6-elongase and delta5-desaturase activities to produce ARA or EPA, respectively, as end-products [24,69]. Further chain lengthening to 22C products (n-6DPA or DHA) uses a delta5-elongase and a delta4-desaturase. Notably, this differs from the ‘Sprecher pathway’ in mammals (see [6,7]). Although the conventional delta6-pathway seems the most common method of producing VLCPUFA in algae, an alternative route (delta8-pathway) has been found in certain species [24]. Moreover, some algae have omega3- (or delta15-) desaturases, that can convert n-6 PUFA to n-3 PUFA [69,76,77].

In some Thraustochytriaceae a polyketide synthase (PKS) can form VLCPUFA. For example, PKS is used by *Schizochytrium* [78], whereas, in *Thraustochytrium*, the desaturase/elongation pathway is utilised (see [24]). Several reviews cover the production of VLCPUFA in algae [66,71,79].

As mentioned above, although LA and ALA are the main fatty acids and are considered the core ‘essential fatty acids’, they are poorly converted to ARA, EPA and DHA, which are the actual metabolic precursors for lipid mediators. Thus, Cunnane has argued for ‘conditional requirements’ of VLCPUFA in the diet [5]. Moreover, the perceived need for more n-3 PUFA in human diets has also led to increased consumption of oily fish (or their oils) as a convenient source of EPA and/or DHA. However, with problems due to over-fishing, as well as the increased demand for more dietary n-3 PUFA [80], this is not sustainable for the future [81,82,83].

The above conundrum has led to a serious examination of algae to supplement the demand for n-3 PUFA, especially EPA and DHA, for human nutrition. As discussed in [48], it is generally thought that ‘many animals’ and ‘most invertebrates’ need to have EPA and DHA in their diet, because of their limited ability to synthesise them from ALA [84]. However, quantitative evidence is currently somewhat lacking [48]. Nevertheless, mammals (like humans) have definite requirements and this has led to extensive use of rodent models for cardiovascular or dementia research [85]. However, availability of n-3 VLCPUFA, especially DHA, is projected to decline globally, due to climate change [54]. The latter causes water temperatures to rise with the consequence that algae contains less VLCPUFA (e.g., [86]).

## 4. Factors Regulating PUFA Formation in Algae

Lipid metabolism in algae is strongly influenced by many environmental factors, which include nutrition (notably N, P and S, and especially their limitation), general conditions (light, pH, temperature) and, of course, toxic materials (e.g., heavy metals) [23,56]. Of interest in the general context of algal utilisation for oils, is that such stresses often cause the accumulation of triacylglycerol (TAG) [24,87].

Clearly, nitrogen is a basic constituent, needed for growth, and minimal requirements have been measured, for example, in *Chlorella vulgaris* [88]. On the other hand, low nitrogen stress will cause accumulation of TAG in green microalgae or diatoms to 20–50% dry weight [24,89]. Naturally, this phenomenon has been mainly studied in algae that offer commercial possibilities, either for biomass/biofuel production or for the biosynthesis of n-3 VLCPUFA. Thus, species such as *P.tricornutum* or *Nannochloropsis* have been studied in detail. The accumulation of TAG is caused by a combination of increased de novo synthesis, as well as transfer of fatty acids from membrane lipids [24]. For *N. gaditana*, the transfer of fatty acids comes from a decline in galactolipids and re-organisation of the photosynthetic apparatus [90]. In agreement, in *P. tricornutum*, the important chloroplast thylakoid constituents monogalactosyldiacylglycerol (MGDG) and phosphatidylglycerol (PtdGro), were reduced considerably, whereas other membrane lipids were largely unaffected [91]. Other details of the effects of N-limitation are covered in [24]. The overall processes can be regulated by a nitrogen response regulator [92], by nitrate reductase expression [93], and by a transcription factor (termed ROC40) [94].

One of the other major nutrients, phosphorus, can also affect lipid metabolism. Clearly, it is required for phospholipid biosynthesis, and P-starvation reduces the amounts of all phosphoglycerides [95]. P-limitation seems to cause a major change in the balance of different lipid classes, and not just phospholipids. For example, the important role of PtdGro in thylakoid functions can be partly replaced by substitution with another anionic lipid, the plant sulpholipid (sulphoquinovosyldiacylglycerol, SQDG). These alterations are described in detail in two reviews [23,24]. In addition, phosphorus deficiency can also cause an accumulation of TAG and, hence, increase algal oil content [65,91,96].

For many diatoms, silicon is a macronutrient, and this includes many oleaginous species [97]. In addition, silicon depletion will increase TAG accumulation in those diatom species that require it [98,99]. *P. tricornutum,* on the other hand, seems to have little requirement for silicon, although this element can benefit growth [24,100].

As mentioned above, there is increasing concern that increases in environmental temperatures (either chronically in the oceans or transiently in streams and ponds) will decrease algal contents of PUFAs and, in particular, VLCPUFAs [54]. Growth in temperature will influence both fatty acid proportions, as well as lipid class composition [23,24,56]. Although many algal cultures exhibit maximal growth in the 20–30 °C range, different species are found in nature to grow at a wide variety of temperatures, ranging from the polar regions to hot springs. Temperature stress can be used to enhance production of valuable metabolites or to alter lipid content of different species [23,24]. General responses to altered ambient temperature in terms of growth, lipid proportions and fatty acid percentages have been described [23,24,56]. In terms of commercial production of algae, changes in lipid quantity and quality are most important. As a generalisation, increased lipid production is often found with temperature stress above the optimal growth values. However, high temperatures usually result in significantly less PUFA proportions. So commercial production may need to consider the main use of the end product. For example, reduced PUFA contents may be desirable if the algae are to be used for biofuel purposes. In contrast, for nutritional applications, lower growth temperatures, which encourage increased PUFA proportions, will be needed. For global PUFA availability [54], temperature stress is very important and specific studies using green microalgae have demonstrated this unequivocally [84,101,102]. Of course, reduced PUFA contents in algae will feed through to higher trophic levels, with implications for invertebrates, for fish and, consequently, human health.

## 5. Algae as Commercial Sources of VLCPUFAs

As discussed above, essential fatty acids (EFA) are needed by mammals and, indeed, many (most?) animals. The VLCPUFAs which are ‘conditionally’ important [5] are produced by algae. Moreover, such acids are needed for good health and a significant dietary deficit may be implicated in many important diseases. Although EFAs (LA, ALA) are provided by higher plant sources, currently very few populations receive adequate n-3 PUFA. Since the enzymes producing lipid mediators are usually active with both n-3 and n-6 precursors (mainly LA and ALA), their ratio in foods is critical. With a dietary ratio of 3:4 for n-6/n-3 PUFA being currently advised [2,103,104], increasing supplies of n-3 PUFA, especially EPA and DHA, are needed. Thus, the commercial requirements for such acids, especially as fish feed in aquaculture, has increased.

The primary producers of PUFAs are photosynthetic organisms, with algae as the main source of EPA and DHA [105]. With the obvious limitation in future fish supplies, commercial usage of algae has increased. Oils from *Crypthecodinium cohnii* (‘DHASCO’) [106] and from *Schizochytrium* spp. [57] are commercially successful, especially for infant nutrition. Algal oils have some advantages over fish oils. They are usually enriched in a single n-3 VLCPUFA and will be devoid of potentially toxic compounds that may be present in fish [57]. Furthermore, because fish are often poor at converting ALA (or LA) into VLCPUFA [107], fish oils are used extensively in aquacultures. Currently, around 75% of marine fish oils are used in aquaculture [108] and this has led to the increasing use of algae as sources of VLCPUFA in fish feeds [109,110]. Several algal species are commercially viable as sources of n-3 VLCPUFA. *Nannochloropsis* spp. and *P. tricornutum* can accumulate EPA to about 40% of their total fatty acids [111,112]. Sources of DHA, with 30–40% of total fatty acids, are *Thraustochytrium* or *Schizochytrium* spp. [113].

The production of EPA by microorganisms and factors affecting its production have been summarised [114]. As noted above, *Nannochloropsis* spp. (e.g., *N. gaditana*, *N. oculata*) and *P. tricornutum* have been well researched [115,116], but other species, such as *Trachydiscus minutus* [117], are being considered. As with most algae, N and P supplies can influence lipid accumulation in the latter alga [118]. There has been some work on the genetic modification of *P. tricornutum* [119] and other algae in order to increase productivity [120]. Indeed, knowledge of the basic biochemistry of *Chlamydomonas* can be applied to commercial algae such as *Nannochloropsis* [24]. Exploitation of algae for EPA production has included not only *Nannochloropsis* [121], but also other species [121,122], such as *Odontella aurita* [123].

For DHA, the first commercial single cell oil was from a dinoflagellate, *C. cohnii*. There are more than 2000 dinoflagellates that have been identified, of which about half can grow without light. DHA accumulation in *C.cohnii* has been thoroughly discussed [122] and the use of different substrates described (see [24]). *Schizochytrium*, together with *C. cohnii*, has been used for over a decade in the infant milk formula market [122]. Commercial production of DHA by *Schizochytrium* is described in [124]. These thraustochytrids synthesise DHA using a PKS system [76], rather than the usual desaturation/elongation pathway. The possible manipulation of algae, including *Schizochytrium*, to produce more n-3 VLCPUFA has been reported [125]. In addition, the use of *Schizochytrium* oils for food or drink additives has been evaluated [124].

Farmed seaweeds are also important and, in 2016, approached 28 million tonnes (wet weight) [126]. Indeed, farmed seaweeds represent 96% of the total global seaweed supply [126]. This farming is mainly in Asia, where six countries produce 98% of the total supply. Indeed, in the Philippines some 70% of the total aquaculture production is aquatic plants, including seaweeds, making the country the second largest producer [127].

Overall reviews of VLCPUFA production by algae have been made [66,71,79,128]. Not only are such acids increasingly used for human nutrition, including in the modification of eggs, meat and milk, but they are also used in pet foods [129] and, especially, in aquaculture [122,130,131,132].

## 6. Conclusions

The importance of VLCPUFA in global ecosystems has been emphasised recently, partly due to climate change, but often driven by the important role of n-3 VLCPUFA for human health. Although fish oils are the main sources of dietary EPA and DHA for humans, these acids are produced de novo by algae. Thus, the biosynthesis of n-3 VLCPUFAs is being increasingly studied in algae, especially commercial species. In turn, algal enzymes used for VLCPUFA formation have been used in transgenic crops. By these means, algae are critical for the future production of EPA and DHA and, thus, for the sustainability of global ecosystems.

## Figures and Tables

**Figure 1 biomolecules-09-00708-f001:**
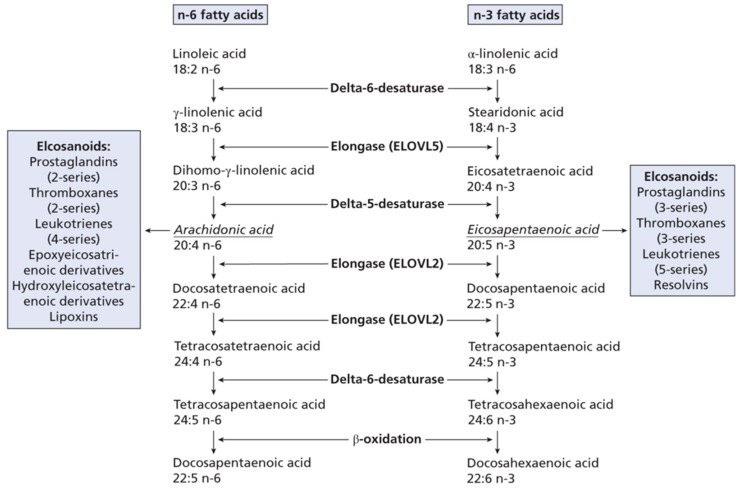
n-3 and n-6 fatty acid metabolism. Taken from [1], with permission. *Lipids: Biochemistry, Biotechnology and Health*, 6th ed. Gurr, M.I.; Harwood, J.L.; Frayn, K.N.; Murphy, D.J.; Michell, R.H. Wiley/Blackwell: Oxford, UK, 2016.

**Figure 2 biomolecules-09-00708-f002:**
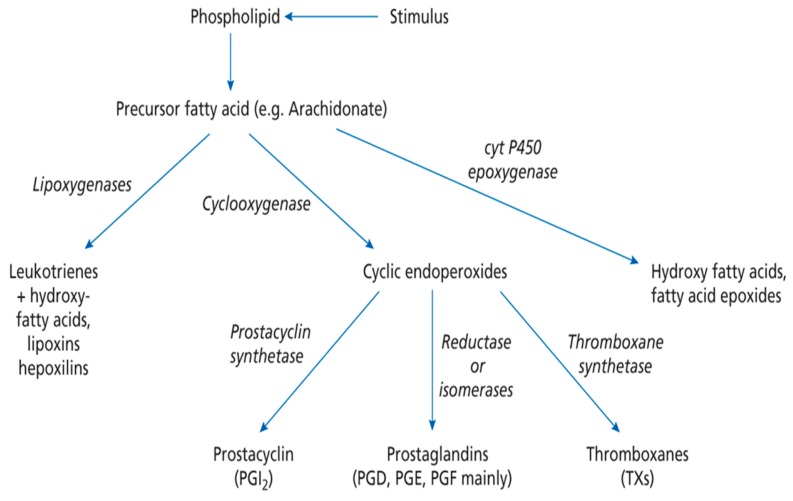
Overall pathway for the conversion of essential fatty acids into eicosanoids. Taken from [1], with permission. *Lipids: Biochemistry, Biotechnology and Health*, 6th ed. Gurr, M.I.; Harwood, J.L.; Frayn, K.N.; Murphy, D.J.; Michell, R.H. Wiley/Blackwell: Oxford, UK, 2016.

**Table 1 biomolecules-09-00708-t001:** Comparison of the fatty acid composition of some commercial fish oils.

	Fatty Acid (% Total)								
	14:0	16:0	16:1	18:1	20:1	20:5	22:1	22:5	22:6
Anchovy	9	17	13	10	1	22	1	2	9
Capelin	7	10	10	14	17	8	15	-	6
Cod Liver	4	10	8	25	10	10	7	1	10
Menhaden	9	19	12	11	1	14	-	2	8
Salmon *	5	12	6	20	10	7	9	3	11
Sardine	8	18	10	13	4	16	3	2	9
Tuna	3	22	3	21	1	6	3	2	22

* Farmed salmon; 16:1, Palmitoleic acid; 18:1, oleic acid; 20:1, eicosenoic (gondoic) acid; 20:5, eicosapentaenoic acid; 22:1, docosenoic (erucic) acid; 22:5, docosapentaenoic acid (mainly 7, 10, 13, 16, 19 isomer); 22:6, docosahexaenoic acid (omega-3 isomer); Data taken from Gunstone et al. [46].

**Table 2 biomolecules-09-00708-t002:** Major fatty acids in a variety of freshwater and marine algae (data taken from [55,58]).

	16:0	16:1	18:1	n6-18:2	n3-18:3	n6-20:4	n3-20:5	n3-22:6
*Chlamydomonas reinhardtii*—^1^	20	4	15	-	22	-	-	-
*Dunaliella salina*—^1^	27	-	11	4	36	-	-	-
*Scenedesmus obliquus*—^1^	31	-	7	8	11	-	-	-
*Chlorella vulgaris*—^2^	11	16	3	25	30	-	-	-
*Lauderia borealis*—^3^	12	21	2	1	-	1	3	-
*Phaeodactylum tricornutum*—^3^	19	25	8	6	1	1	18	1
*Nannochloropsis gaditana*—^4^	15	30	5	-	9	4	35	-
*Emilia huxleyi*—^5^	19	10	20	-	-	-	-	9
*Pavlova lutheri*—^5^	20	26	2	1	1	-	18	10
*Ectocarpus siliculosus*—^6^	15	-	-	6	30	10	13	-
*Fucus vesiculosus*—^6^	21	2	26	10	7	15	8	-
*Chondrus crispus*—^7^	34	6	9	1	1	18	22	-
*Porphyridium purpureum*—^7^	25	-	-	23	-	39	13	-

1—Chlorophyceae; 2—Trebouxiaphyceae; 3—Bacillariophyceae; 4—Eustigmatophyceae; 5—Haptophyceae; 6—Phaeophyceae; 7—Rhodophyceae. Fatty acid identities as for Table 1; 18:2, linoleic acid; 18:3, alpha-linolenic acid. For extra information, see [23,24,55,56,58]

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
