# Peer review of "Algae: Critical Sources of Very Long-Chain Polyunsaturated Fatty Acids"

_biomolecules, 2019, doi:10.3390/biom9110708_

Round 1

Reviewer 1 Report

Very interesting and useful review which updates information on production and application of w3 fatty acids isolated from algae. Special attention paid to production of “conditionally essential” eicosapentaenoic and docosahexaenoic acids, which are direct precursors of eicosanoids and other active oxidized derivatives and superior than LA and ALA. In this article discussed different factors acting on PUFA synthesis which are important for commercial success of algae oil production. Taking in mind total decline of fish catch and respective fish oil shortage, it possible forecast increase of algae oil significance.

I suggest change formula on line 43 

line 145 EM or ER&

Author Response

Comment added about the shortage of VLCPUFAs in the future and corrections made.

Reviewer 2 Report

The manuscript reviews the basic biochemistry of polyunsaturated fatty acid (PUFA) production by microalgae, and summarises their role as both dietary and commercial sources of very long-chain PUFA, particularly involved in aquaculture. This is contrasted against marine fish as sources of very long-chain PUFA for similar applications. Physiological aspects of microalgal PUFA production are also summarised. The review manuscript gives a comprehensive summary of literature in the field, and should be a valuable resource for researchers to access specific studies to enable further research.

Major Comments:

Figure 1. Second column (n-3 fatty acids). The short-hand notation for Docosapentaenoic acid requires correction. Should read “22:5n-3” not “22:6n3”. Table 1 & 2: The author may wish to consider inclusion of (n-3) / (n-6) notation of the PUFAs in the column headings, as this would aid the unfamiliar reader with more complete interpretation of the data. While the data presented also perfectly illustrates the author’s objective, he may wish to also consider reference to additional contemporary sources for the reader’s benefit. Lines 111-113 & 167-169: The author discusses the need for further PUFA sources to reduce the current and future projected shortfalls in VLCPUFA availability. It would benefit the reader if this point could be expanded with some current estimates of these factors. Line 163: may the author consider addition of the term “in animals”(or similar) after the phrase “converted to ARA, EPA and DHA”.

Minor Comments:

Line 16, 60, 66: space is required before open bracket. Line 20, 51: space is required before citations. Line 44, 154: citation last bracket requires correction. Line 59: correction “and” and “ad”. Line 66: space required after end of sentence. Line 75: delete period. Line 247: citation requires comma separating numbers, not period. Please delete “and” after citations. Line 272: citation requires comma separating numbers, not period. Line 280: delete space from within citation.

Author Response

Figure 1 corrected.

Table 2 amended as suggested. Table 1 was not changer because to add n-3 and n-6 would have been misleading and the legend provides the details.

All other amendments made as suggested except line 163 (see previous explanation---Cunnane refers only to humans and the following paragraph covers the point about other animals).

Cunnane's paper was concerned with humans. The next paragraph mentions other animals so I have left the text.

Reviewer 3 Report

This manuscript reports a comprehensive overview, timely and well written review focused on the biosynthesis of polyunsaturated fatty acids (PUFAs) from algae together with the main factors involved in the regulation of their biosynthetic process. Moreover, this review emphasise the importance of algae as sustainable sources of PUFAs highly important for human nutrition and health by aiding the maintenance of the physiological functions, but also as a feed thus improving the nutritional value of fish farmed in aquaculture.

There are only some small issues that need to be addressed as will be detailed. Based on this and in the following items, I recommend this manuscript for publication with minor revisions.

General Comments:

The manuscript is focused on algae as key sources of very long chain (VLC)-PUFAs. Some examples of algae (macro and microalgae) having high content in VLC-PUFAs are presented. However, Palmaria palmata (Rhodophyceae) which is a very interesting red macroalgae and has a high content in the important n-3 VLC-PUFA eicosapentaenoic acid (EPA, C20:5), near to 50% of fatty acid content is not referred in this manuscript. Moreover, this is an edible seaweed with a long tradition as food supplement consumed in several countries.

Please add this information in the main text and also some references, for example as following:

Vincent JT van Ginneken et al. Polyunsaturated fatty acids in various macroalgal species from north Atlantic and tropical seas. Lipids Health Dis. 2011; 10: 104.

Joe Fleurence et al. Fatty acids from 11 marine macroalgae of the French Brittany coast. Journal of Applied Phycology 6: 527-532, 1994.

K. Mishra et al. Lipids of the Red Alga, Palmaria palmate. Botanica Marina. Vol. 36, pp. 169-174, 1993.

Britta Grote. Recent developments in aquaculture of Palmaria palmate (Linnaeus) (Weber & Mohr 1805): cultivation and uses. Reviews in Aquaculture (2019) 11, 25–41.

In this way, it will be interesting to make some statements on this promising macroalgae.

In page 7, in table 2 the values are expressed as % of total or esterified fatty acids (namely in betaine, phospholipids, and glycolipids)? This information is not clear and should be revised. The PUFAs in algae are usually esterified in polar lipids namely in phospholipids and glycolipids.

Nevertheless, the information presented in this table is quite useful and it will be of high interest. In this way, it would be very interesting if this table could be completed with some more information on fatty acid composition from other marine algae.

Another point that should be interesting to explore a little in deep is the negative impact of climatic changes in the composition of algae and consequently in their fatty acid profile. Algae plasticity allows them to adapt their lipid profile under certain grown conditions. Under the present climacteric conditions, this adaptability can negatively impact the algae nutritional value. In this way, it would be interesting to refer some alternative and already on-going strategies that may help to maintain algae biochemical composition and thus it the healthy and nutritional properties of algae. For example, farmed seaweed production it was reported to represent 95.6% of the global seaweed supply (Chopin, T. (2018a). Seaweed aquaculture – From the global, mostly Asian, picture to the opportunities and constraints of the Canadian scene. Bull. Aquacult. Assoc. Canada 2017-1, 3–8). Some research work that has been performed using seaweed farmed in integrated multi-trophic aquaculture systems has demonstrated that seaweed grown under that controlled environments are still a sustainable source of PUFAs, also contributing for  the biomass stabilization and to the valorization and commercial value of farmed algae.

Specific comments:

Page 1, line 34 – “cyclooxygenase” should be “cyclooxygenase”

Page 5, line 75 – end point after reference [5,31] should be removed

Page 11, line 280 – there is a space in reference 76. Remove or confirm if a difference reference is supposed to be there.

Page 15, reference 50 – Dapnhia should be in italic

Page 15, reference 59 – “Grateloupia turutura” shoud be “Grateloupia turuturu”: Non-methylene Interrupted and Hydroxy Fatty Acids in Polar Lipids of the Alga Grateloupia turuturu Over the Four Seasons

Page 16, reference 71 – “Chlamydomonas” should be in italic

Page 19, reference 108 – “Sparus aurata” should be in italic

Author Response

Comments added about Palmaria palmata with new references.

Comments added about the use of seaweeds, with new references.

All specific comments addressed